# Computational Analysis for Bioconvection of Microorganisms in Prandtl Nanofluid Darcy–Forchheimer Flow across an Inclined Sheet

**DOI:** 10.3390/nano12111791

**Published:** 2022-05-24

**Authors:** Jianfeng Wang, Zead Mustafa, Imran Siddique, Muhammad Ajmal, Mohammed M. M. Jaradat, Saif Ur Rehman, Bagh Ali, Hafiz Muhammad Ali

**Affiliations:** 1College of Mechanical Engineering, Xijing University, Xi’an 710123, China; jidianxijng@163.com; 2Mathematics Program, Department of Mathematics, Statstics and Physics, College of Arts and Sciences, Qatar University, Doha P.O. Box 2713, Qatar; mmjst4@qu.edu.qa; 3Department of Mathematics, University of Management and Technology, Lahore 54770, Pakistan; muhammadajmally485@gmail.com (M.A.); saifurrehman8684@gmail.com (S.U.R.); 4Department of Applied Mathematics, Northwestern Polytechnical University, Xi’an 710129, China; baghalisewag@mail.nwpu.edu.cn; 5Faculty of Computer Science and Information Technology, Superior University, Lahore 54000, Pakistan; 6Mechanical Engineering Department, King Fahd University of Petroleum & Minerals, Dhahran 31261, Saudi Arabia; 7Interdisciplinary Research Center for Renewable Energy and Power Systems (IRC-REPS), King Fahd University of Petroleum and Minerals, Dhahran 31261, Saudi Arabia

**Keywords:** Prandtl nanofluid, bioconvection, magnetohydrodynamic, stratification, inclined sheet

## Abstract

The two-dimensional boundary layer flow of a Prandtl nanofluid was explored in the presence of an aligned magnetic field over an inclined stretching/shrinking sheet in a non-Darcy permeable medium. To transform the PDEs of the leading equations into ODEs, a coupled boundary value problem was formed and suitable similarity functions were used. To obtain numerical answers, an efficient code for the Runge–Kutta technique with a shooting tool was constructed with a MATLAB script. This procedure is widely used for the solution of such problems as it is efficient and cost-effective with a fifth-order accuracy. The significance of immersed parameters on the velocity, temperature, concentration, and bioconvection is shown through figures. Furthermore, the physical parameters of the skin friction coefficient and the Nusselt numbers are demonstrated in tables. The declining behavior of the flow velocity against the porosity parameter $K_{p}$ and the local inertia co-efficient Fr is shown, and the both parameters of the Darcy resistance and Darcy–Forchheimer resistance are responsible for slowing the fluid speed. The increasing values of the Schmidt number Sc decrease the concentration of the nano entities.

## 1. Introduction

Microorganisms travel randomly in a single-cell or colony-like configuration during bioconvection, which is a natural phenomenon. Bioconvection systems are based on the directed motion of several types of microorganisms. In still water, gyrotactic microorganisms travel upstream against gravity, causing the suspensions to be thicker at the top than at the bottom. Bioconvection has numerous applications in biotechnology and industry, such as organic microsystems, ethanol, biofuel, and fertilizers. In oil refineries, bioconvection is also quite essential. In enzyme biosensors, bioconvection is employed. Because of these intriguing traits, several academics have analyzed several applications. Alshormani et al. [1] discussed the numerical study of the stream of the bioconvection of a viscoelastic nanofluid. Chu et al. [2] studied the importance of magnetohydrodynamic bioconvection and energy in a stream of non-Newtonian fluid towards an extended surface—a Buongiorno research model. Dey et al. [3] analyzed the stream of a dusty nanofluid in the presence of a vertically extending surface. Ferdows et al. [4] improved the thermal transportation of the nanofluid and stream in MHD bioconvection through a significantly stretchable sheet. Irfan et al. [5] conducted a time-dependent MHD bionanofluid stream in a permeable medium with thermal radiation close to a shrinking sheet. Khan et al. [6] investigated the entropy creation in a stream of nanofluid undergoing bioconvection joining two stretchable turning disks. Abbasi et al. [7] discussed the bioconvection stream of a viscoelastic nanofluid past a convective turning stretching disk. By using the power law, Asjad et al. [8] explored fractional bioconvection over a vertical sheet. Oyelakin et al. [9] presented the nonlinear thermal radiation and various transport properties that influence the two-dimensional movement of a Casson nanofluid containing gyrotactic microorganisms across a rotating wedge.

Nanofluids are developed by suspending nanoparticles in common carrier liquids such as oil, argon, or ethylene glycol. Because of its expanding applicability in a variety of sectors, such as nano-drug delivery and sophisticated nuclear systems, nanofluids have been widely employed by multiple scientists. These fluids are also utilized for cooling in a variety of heat transfer equipment, such as electronic freezing systems and heat exchangers. Choi [10] was the first to explore nanofluids. Hayat et al. [11] introduced a second-grade nanofluid flow over a convected thermal extending sheet by the impacts of a magnetic field. Hayat et al. [12] studied the entropy creation of the minimization through the curved extending surface with partial slip by a Darcy–Forchheimer nanofluid stream. Sudarana et al. [13] studied the applications of a nanofluid stream above a permeable extending sheet having heat radiation with the effects of the double stratification of the transport of heat and mass and chemical reactions. Sreedevi et al. [14] analyzed the time-dependent hybrid nanofluid stream across an extended sheet having heat radiation with the effect of thermal and mass transport. Ali et al. [15] presented a finite-element method approach on a time-dependent MHD axisymmetric nanofluid flow past an extended layer by thermo-diffusion. Shoaib et al. [16] analyzed a nanofluid in the presence of a heat radiative extended surface employing the numerical computation of a rotating stream. Ghosh et al. [17] analyzed the Casson nanofluid flow of viscoelastic bioconvection with a magnetic field and Cattaneo–Christov double diffusion. More work on nanofluids of various types of geometries was given in [18,19,20]. Very recently, a wide range of applications of non-Newtonian fluid has motivated researchers to examine non-Newtonian fluids. These are utilized in the production mechanisms in industrial fields, the production of petroleum, and a broad range of chemical reactions in the chemical engineering field. Many researchers have examined non-Newtonian fluids through different types of geometries [21,22,23].

Magnetohydrodynamic (MHD) flow is very important in the production of petroleum and the metallurgical process. It is worth noting that the ultimate product created is impacted by the rate at which these processes cool. The magnetic field is utilized to purify molten metals to separate metallic materials from nonmetallic components. Sarpakaya [24] was the first to examine the MHD flow in this particular scenario. Magnetohydrodynamics (MHD) has applications in medicine, astronomy, sophisticated plane design, and effectively dealing with thermal transportation rates in cylinders, as well as a variety of machines, including turbulent pumps, and energy producers. Nagaraja et al. [25] analyzed the convective stream of MHD Casson fluid due to a chemical reaction through a curved extending sheet in the presence of an exponentially space-dependent thermal source. Daniel et al. [26] discussed the mixed convection stream of a time-dependent MHD nanofluid past an extending sheet, as well as the effects of the slip condition, electric field, and thermal radiation. Khan et al. [27] studied a time-dependent MHD viscoelastic nanofluid flow past a permeable extending sheet in the presence of radiation and the effects of multiple slips using the finite-element technique. Shateyi et al. [28] investigated in detail the MHD time-dependent boundary surface flow of a Williamson fluid past an extending sheet with the numerical computation of thermal and mass transport. Yasmin et al. [29] presented a micropolar fluid over an extending surface with the mass and thermal transport in MHD flow. Jabeen et al. [30] studied permeable media with thermophoresis, chemical reaction, and radiation through a linearly extending sheet in the investigation of MHD flow. Ibrahim et al. [31] introduced the flow of an MHD nanofluid due to a chemical reaction with an upper convected extending surface. Nayak et al. [32] studied the entropy-optimized Darcy–Forchheimer flow on the interfacial layer and the shape impacts of the modified Hamilton–Crosser model. Nayak et al. [33] discussed the MHD microrotations of cross nanomaterials having a cubic autocatalytic chemical reaction with the entropy minimized.

The study of boundary-layer streams over stretching surfaces has gone well beyond numerous practical applications in manufacturing, industrial processes, metallurgical, and engineering approaches. Paper making, glass blowing, crystal growth, annealing, and tinning copper wire are all examples of applications of this kind. Much research has recently been conducted on diverse fluid streams passing via stretching sheets. Crane [34] investigated the boundary layer flow of a Newtonian fluid induced by the stretching sheet moving in its plane with the velocity fluctuating linearly with the distance from a fixed point owing to consistent stress application. Cortell et al. [35] studied the heat transfer and viscous stream past a nonlinearly extending surface. Gangadhar et al. [36] discussed the stream of nanofluids to investigate the thermal absorption/generation in the presence of variable suction/injunction past a stretching sheet with viscous dissipation. Razzaghi et al. [37] conducted the numerical simulation of a stream past a nonlinearly extending sheet considering the magnetic field and a chemical reaction. Rehman et al. [38] studied the distribution of the stream on the mass and heat transport of an MHD thin fluid film past a time-dependent stretching sheet with variational physical effects. Pal et al. [39] described the mass and heat transport of a non-Newtonian Jeffery nanofluid past an extruded extended sheet in the presence of a nonuniform thermal source/sink and heat radiation. Kairi et al. [40] studied the bioconvection of gyrotactic microorganisms on thermosolutal Marangoni convection across an inclined surface. Mkhatshwa et al. [41] discussed the bioconvective Casson nanofluid flow on an MHD radiative chemical reaction across a vertical surface.

Motivated by the above-cited literature and their recent applications in modern technology, the core objective of the present study was the computational analysis of the bioconvection of microorganisms in a Prandtl nanofluid’s Darcy–Forchheimer flow across an inclined sheet. The inspiration of this work was to improve the thermal transportation of the base fluid having a mild inclusion of nano species. This was considered to meet the rising need for the thermal management of the modern, complicated equipment and gadgets used in everyday life and thermal engineering. To improve the base fluid’s stability and avoid nanoparticle sedimentation, we incorporated microorganisms, which were very useful to improve the stability. Furthermore, we utilized a stratified medium, and no such study has been performed to the best of our knowledge until now. Fluid motion and heat transport over an inclined sheet are rarely discussed with nano inclusion and bioconvection, and we sought to fill this existing gap in the literature. The findings are focused on being utilized in heat exchanges of various types.

## 2. Mathematical Formulation

A computational analysis of the flow of the bioconvection of microorganisms in a Prandtl nanofluid over an inclined surface with an inclination angle α that is being stretched along the x−axis with stretching velocity u${}_{w}=ax$ is described in this model. The base fluid contains a modest diffusion of nanoparticles and microorganisms. The magnetic field B${}_{0}$ acts in the *y*-direction. The temperature at the wall is ${\tilde{T}}_{f}={\tilde{T}}_{0}+a_{1}x$; the concentration at the wall is ${\tilde{C}}_{f}={\tilde{C}}_{0}+a_{2}x$; the motile microorganisms is N${}_{f}={\tilde{N}}_{0}+a_{3}x$ and conserved at the wall. The physical representation of the model is given in Figure 1. The constitutive equation for the Prandtl fluid was given by Akbar [42].
(1)$$\begin{array}{c} \tau^{*}=\frac{A\sin^{-1}\left(\frac{1}{C}{\left[{\left(\frac{\partial \tilde{u}}{\partial y}\right)}^{2}+{\left(\frac{\partial \tilde{v}}{\partial x}\right)}^{2}\right]}^{0.5}\right)}{{\left[{\left(\frac{\partial \tilde{u}}{\partial y}\right)}^{2}+{\left(\frac{\partial \tilde{v}}{\partial x}\right)}^{2}\right]}^{0.5}} \end{array}$$

In the above equation, *A* and *C* are the material constants of the Prandtl fluid model. The governing equations of the continuity, momentum, temperature, concentration, and bioconvection of the fluid flow are given as [43,44,45]:(2)$$\begin{array}{c} \frac{\partial \tilde{u}}{\partial x}+\frac{\partial \tilde{v}}{\partial y}=0, \end{array}$$
(3)$$\begin{array}{c} \left(\tilde{u}\frac{\partial \tilde{u}}{\partial x}+\tilde{v}\frac{\partial \tilde{u}}{\partial y}\right)=\frac{A\nu }{C}\left(\frac{\partial^{2}\hat{u}}{\partial y^{2}}\right)+\frac{A\nu }{2C^{3}}{\left(\frac{\partial \tilde{u}}{\partial y}\right)}^{2}\left(\frac{\partial^{2}\tilde{u}}{\partial y^{2}}\right)-\frac{\sigma B^{2}\left(x\right)}{\rho }Sin^{2}\gamma \tilde{u}\\ -\frac{\nu }{k}\tilde{u}-F{\tilde{u}}^{2}+g\left[\beta_{1}\rho_{f}\left(1-\tilde{C_{\infty }}\right)\left(\tilde{T}-\tilde{T_{\infty }}\right)-\left(\rho_{p}-\rho_{f}\right)\left(\tilde{C}-\tilde{C_{\infty }}\right)-\gamma \left(\rho_{m}-\rho_{f}\right)\left(\tilde{N}-\tilde{N_{\infty }}\right)\right]Cos\alpha , \end{array}$$
(4)$$\begin{array}{c} \tilde{u}\frac{\partial \tilde{T}}{\partial x}+\tilde{v}\frac{\partial \tilde{T}}{\partial y}=\frac{K}{\rho c_{p}}\frac{\partial^{2}T}{\partial y^{2}}+\tau \left[D_{B}\frac{\partial \tilde{c}}{\partial y}+\frac{D_{T}}{T_{\infty }}{\left(\frac{\partial \tilde{T}}{\partial y}\right)}^{2}\right], \end{array}$$
(5)$$\begin{array}{c} \tilde{u}\frac{\partial \tilde{C}}{\partial x}+\tilde{v}\frac{\partial \tilde{C}}{\partial y}-D_{B}\frac{\partial^{2}\tilde{C}}{\partial {\tilde{y}}^{2}}=\frac{D_{T}}{T_{\infty }}\frac{\partial^{2}\tilde{T}}{\partial y^{2}}, \end{array}$$
(6)$$\begin{array}{c} \tilde{u}\frac{\partial \tilde{N}}{\partial x}+\tilde{v}\frac{\partial \tilde{N}}{\partial y}=\frac{dw_{c}}{C_{f}-C_{0}}\frac{\partial }{\partial y}\left(\tilde{N}\;\frac{\partial \tilde{C}}{\partial y}\right)+D_{N}\left(\frac{\partial^{2}\tilde{N}}{\partial y^{2}}\right). \end{array}$$

The velocity components $\tilde{u}$ and $\tilde{v}$ are along the x−axis and y−axis, respectively; *k* is the porous medium permeability; $F=\frac{C_{b}}{\sqrt{k}}$ represents the inertia coefficient of the porous material; *g* is the gravitational acceleration; $\rho_{p},\rho_{f}$ are the mass density and the fluid density; $C_{b}$ symbolizes the drag coefficient; $\tilde{T}$ represents the temperature of the fluid; τ is the ratio of the heat capacities of the fluid and nanoparticles; C symbolizes the nanoparticles’ concentration; *N* indicates the density of the motile microorganisms; $D_{T}$, $D_{N}$, and $D_{B}$ represent the thermophoretic diffusion coefficient, the diffusivity of the microorganisms, and the Brownian diffusion coefficient, respectively; $T_{f}$, $C_{f}$, and $N_{f}$ characterize the temperature, concentration, and density of the motile microorganisms, at the wall; $T_{\infty }$, $C_{\infty }$, and $N_{\infty }$ are the corresponding ambient values, respectively. The following are the dimensional boundary conditions [43]:(7)$$\begin{array}{c} \left.\begin{array}{c} \tilde{u}=\epsilon u_{w}=ax,\tilde{v}=v_{0},\;\tilde{T}=\tilde{T_{f}}=\tilde{T_{0}}+a_{1}x,\;\tilde{C}=\tilde{C_{f}}=\tilde{C_{0}}+a_{2}x,\tilde{N}=\tilde{N_{f}}=\tilde{N_{0}}+a_{3}x,\;at\;y=0,\\ \tilde{u}\to 0,\;\tilde{T}\to \infty =\tilde{T_{0}}+d_{1}x,\;\tilde{C}\to \infty =\tilde{C_{0}}+d_{2}x,\tilde{N}\to \infty =\tilde{N_{0}}+d_{3}x\;as\;y\to \infty .\; \end{array}\right\} \end{array}$$

To convert the PDEs into a system of ODEs, we employed the similarity transforms below [43]:(8)$$\begin{array}{c} \left.\begin{array}{ccc} \psi =\sqrt{\tilde{v}xu_{w}\left(x\right)}f\left(\eta \right)=\sqrt{a\tilde{v}}xf\left(\eta \right),\eta & = & \sqrt{\frac{{}_{{\tilde{u}}_{w}\left(x\right)}}{\tilde{v}x}y}={\left(\frac{a}{\tilde{v}}\right)}^{\frac{1}{2}}y,\\ \theta \left(\eta \right)=\frac{\tilde{T}-{\tilde{T}}_{\infty }}{{\tilde{T}}_{f}-{\tilde{T}}_{0}},\phi \left(\eta \right)=\frac{\tilde{C}-{\tilde{C}}_{\infty }}{{\tilde{C}}_{f}-{\tilde{C}}_{0}},\chi \left(\eta \right)=\frac{\tilde{N}-{\tilde{N}}_{\infty }}{{\tilde{N}}_{f}-{\tilde{N}}_{0}},\; \end{array}\right\} \end{array}$$
ψ is the stream function. $\tilde{u}=\frac{\partial \psi }{\partial y}$, and $\tilde{v}=-\frac{\partial \psi }{\partial x}$. Equation (Equation 1) is fulfilled in view of the above appropriate relations, and Equations (3)–(7) become:(9)$$f^{{}^{{}^{\prime \prime \prime }}}\delta \left(1+\beta f^{{}^{\prime \prime }2}\right)+f^{{}^{{}^{\prime \prime }}}f-M\sin^{2}\gamma f^{{}^{\prime }}-K_{p}f^{{}^{\prime }}-\left[1+Fr\right]f^{{}^{\prime }2}+\omega \left[\theta -Nr\phi -Rb\chi \right]\cos\alpha =0,$$
(10)$$\frac{1}{\mathrm{Pr}}\theta^{{}^{\prime \prime }}-S_{1}{\tilde{f}}^{{}^{\prime }}-f^{{}^{\prime }}\theta +f\theta^{{}^{\prime }}+Nb\theta^{{}^{\prime }}\phi^{{}^{\prime }}+Nt\theta^{{}^{\prime }2}=0,$$
(11)$$\phi^{{}^{\prime \prime }}-Sc\left[S_{2}f^{{}^{\prime }}+f^{{}^{\prime }}\phi -f\phi^{{}^{\prime }}\right]+\frac{N_{t}}{N_{b}}\theta^{{}^{\prime \prime }}=0,$$
(12)$$\chi^{{}^{\prime \prime }}-LbPr\left[S_{3}f^{{}^{\prime }}+\chi f^{{}^{\prime }}-\chi^{{}^{\prime }}f\right]-Pe\left[\Omega \phi^{{}^{\prime \prime }}+\chi \phi^{{}^{\prime \prime }}+\chi^{{}^{\prime }}\phi^{{}^{\prime }}\right]=0,$$
having dimensionless boundary constraints,
(13)$$\begin{array}{c} \left.\begin{array}{c} f\left(\eta \right)=S,\;f^{\prime }\left(\eta \right)=\epsilon ,\;\theta \left(\eta \right)=1-S_{1},\;\phi \left(\eta \right)=1-S_{2},\chi \left(\eta \right)=1-S_{3},\;at\;\eta =0,\\ f^{\prime }\left(\eta \right)\to 0,\theta \left(\eta \right)\to 0,\phi \left(\eta \right)\to 0,\chi \left(\eta \right)\to 0,\;\;as\;\eta \to \infty .\; \end{array}\right\} \end{array}$$
where δ=
$\frac{A}{C}$ and $\beta =\frac{a^{3}x^{2}}{\upsilon }$ are the fluid parameters, $Nr=\frac{\left(\rho_{p}-\rho_{f}\right)\left({\tilde{C}}_{f}-{\tilde{C}}_{0}\right)}{\beta \rho_{f}\left(1-c_{\infty }\right)\left({\tilde{T}}_{f}-{\tilde{T}}_{0}\right)}$ signifies the buoyancy parameter, M=
$\frac{\sigma \beta_{0}^{2}}{\rho }$ indicates the magnetic parameter, $K_{p}=\frac{\upsilon }{ka}$ denotes the porosity parameter, $Rb=\frac{\left(\rho_{m}-\rho_{f}\right)\left({\tilde{N}}_{f}-{\tilde{N}}_{0}\right)}{\beta_{1}\rho_{f}\left(1-c_{\infty }\right)\left({\tilde{T}}_{f}-{\tilde{T}}_{0}\right)}$ represents the bioconvection Rayleigh number, $Fr=\frac{C_{b}}{\sqrt{K}}x$ indicates the local inertia coefficient, $S_{1}=\frac{d_{1}}{a_{1}},S_{2}=\frac{d_{2}}{a_{2}}$, $S_{3}=\frac{d_{3}}{a_{3}}$ symbolize the thermal stratification parameters, respectively, *S* is the suction/injection parameter, $\omega =\frac{g\beta_{1}\rho_{f}\left(1-c_{\infty }\right)\left({\tilde{T}}_{f}-{\tilde{T}}_{0}\right)}{u_{w}^{2}\left(x\right)}$ is the mixed convection, $Pr=\frac{\rho c_{p}\upsilon }{k}$ is the Prandtl number, $Sc=\frac{k}{D_{B}}$ signifies the Schmidt number, $Nb=\frac{\tau D_{B}\left({\tilde{C}}_{f}-{\tilde{C}}_{0}\right)}{\upsilon }$ represents the Brownian parameter, $Nt=\frac{\tau D_{T}\left({\tilde{T}}_{f}-{\tilde{T}}_{0}\right)}{{\tilde{T}}_{\infty }v}$ denotes the thermophoresis parameter, $Sc=\frac{\nu }{D_{B}}$ indicates the Schmidt number, $Lb=\frac{\upsilon }{D_{N}}$ is the Lewis number, *Pe* = $\frac{dw_{c}}{D_{N}}$ indicates the Peclet number, and $\Omega =\frac{{\tilde{N}}_{\infty }}{{\tilde{N}}_{f}-{\tilde{N}}_{0}}$ indicates the density ratio of the motile microorganisms.

## 3. Physical Quantities

This segment describes the attributes of important physical quantities of engineering interest. The local skin friction $C_{fx}$, motile density number $Nn_{x}$, Sherwood number $Sh_{x}$, and Nusselt number $Nu_{x}$ as given below
(14)$$\begin{array}{c} C_{fx}=\frac{2\tau_{w}}{\rho_{f}u_{w}^{2}\left(x\right)}, \end{array}$$
(15)$$\begin{array}{c} Nu_{x}=\frac{xq_{w}}{k({\tilde{T}}_{w}-\tilde{T_{\infty }})}, \end{array}$$
(16)$$\begin{array}{c} Sh_{x}=\frac{xq_{m}}{D_{B}\left(\tilde{C}-\tilde{C_{\infty }}\right)}, \end{array}$$
(17)$$\begin{array}{c} Nn_{x}=\frac{xq_{n}}{D_{B}\left(\tilde{N}-\tilde{N_{\infty }}\right)}; \end{array}$$
$\tau_{w}$, $q_{w}$, $q_{m}$, and $q_{n}$ are the shear stress, heat flux, mass flux, and motile microorganisms flux, respectively, given as
(18)$$\begin{array}{c} \tau_{w}={\left(\frac{A}{C}\frac{\partial \tilde{u}}{\partial y}+\frac{A}{6C^{3}}{\left(\frac{\partial \tilde{u}}{\partial y}\right)}^{3}\right)}_{y=0}, \end{array}$$
(19)$$\begin{array}{c} q_{w}=-k{\left(\frac{\partial \tilde{T}}{\partial y}\right)}_{y=0},\; \end{array}$$
(20)$$\begin{array}{c} q_{m}=-D_{B}{\left(\frac{\partial \tilde{C}}{\partial y}\right)}_{y=0},\; \end{array}$$
(21)$$\begin{array}{c} q_{n}=-D_{N}{\left(\frac{\partial \tilde{N}}{\partial y}\right)}_{y=0}.\; \end{array}$$
The local skin friction coefficient $C_{fx}$, Nusselt number $Nu_{x}$, Sherwood number $Sh_{x}$, and density number of motile microorganisms $Nn_{x}$, given as
(22)$$\begin{array}{c} Re_{x}^{\frac{1}{2}}C_{fx}=-\left(\frac{\delta \beta }{3}f^{\prime \prime 3}\left(0\right)+\delta f^{\prime \prime }\left(0\right)\right), \end{array}$$
(23)$$\begin{array}{c} Re_{x}^{\frac{-1}{2}}Nu_{x}=\left(-\theta^{\prime }\left(0\right)\right),Re_{x}^{\frac{-1}{2}}Sh_{x}=\left(-\phi^{\prime }\left(0\right)\right),Re_{x}^{\frac{-1}{2}}Nn_{x}=\left(-\chi^{\prime }\left(0\right)\right). \end{array}$$

## 4. Solution Procedure

The resultant set of boundary values in Equations (9)–(12) with the initial and boundary constraints (13) cannot be solved analytically. They are solved numerically by using the Runge–Kutta method, with the shooting techniques, coded in MATLAB. The higher-order derivatives involved in the final governing equations are reduced as follows [44,46]$s_{1}^{\prime }=s_{2},$$s_{2}^{\prime }=s_{3},$$s_{3}^{\prime }=\frac{\left(-1\right)}{\delta (1+\beta s_{3})}\left[s_{1}s_{3}-MSin^{2}\gamma \left(s_{2}-\varsigma s_{3}s_{1}\right)-K_{p}s_{1}-\left(1+Fr\right)s_{1}^{2}+\omega \left(s_{4}-Nrs_{6}-Rbs_{8}\right)Cos\alpha \right],$$s_{4}^{\prime }=s_{5},$$s_{5}^{\prime }=\left(Pr\right)\left[S_{1}s_{2}+s_{2}s_{4}-s_{1}s_{5}-Nbs_{5}s_{7}-Nts_{5}^{2}\right],$$s_{6}^{\prime }=s_{7},$$s_{7}^{\prime }=\left(Sc\right)\left[S_{2}s_{2}+s_{2}s_{6}-s_{1}s_{7}\right]-\frac{Nt}{Nb}s_{5}^{\prime },$$s_{8}^{\prime }=s_{9}$$s_{9}^{\prime }=\left[Lb\left(S_{3}s_{2}+s_{8}s_{2}-s_{1}s_{9}\right)+Pe\left(\Omega s_{7}^{\prime }+s_{7}^{\prime }s_{8}+s_{7}s_{9}\right)\right].$

The corresponding boundary conditions are as follows:$$\begin{array}{c} s_{1}=S,\;s_{2}=\epsilon ,\;s_{4}=1-S_{1},\;s_{7}=1-S_{2},\;s_{9}=1-S_{3},\;at\;\eta =0,\;\\ s_{2}\to 0,\;s_{4}\to 0,\;s_{7}\to 0,\;s_{9}\to 0,\;as\;\eta \to \infty .\; \end{array}$$

## 5. Results and Discussion

The finally transformed Equations (9)–(12) constitute a nonlinear boundary value problem. As usual, such equations are difficult to solve analytically. Thus, a numerical approach based on the Runge–Kutta method was employed here to yield the solution of this problem. First of all, the validity of the numerical scheme was established in the limiting case to compare with the existing studies [43,47,48,49] (see Table 1 and Table 2). A close agreement of the two sets (present and previous) of the result provided confidence in the numerical procedure. Rigorous computational effort was made to evaluate the influence of specific numbers on the skin friction $-f^{{}^{\prime \prime }}\left(0\right)$, local heat transfer rate $-\theta^{{}^{\prime }}\left(0\right),$ normalized fluid velocity, temperature, concentration of nanoparticles, and microorganism density functions. The default values of the involved parameters were: $\delta =2.5,\beta =1.0,M=1.0,\gamma =30,K_{p}=0.3,Fr=0.1,\omega =0.1,Nr=0.3,Rb=0.1,\alpha =30,Pr=1.0,S_{1}=0.1,Nb=0.3,Nt=0.3,$
$Sc=1.5,S_{2}=0.1,Lb=0.3,Pe=1.0,\Omega =0.1,S_{3}=0.1,\epsilon =1.5$. In Table 3, the skin friction increasesin magnitude when M,Kp,δ,Fr,Nr, and Rb increase, but $-f^{{}^{\prime \prime }}\left(0\right)$ is reduced against ω. As seen from Table 4, the Nusselt number decreases against Nb, Nt, and $S_{1}$, and it increases in magnitude directly with Pr. The increment of Pe and Lb on the motile density number $Re_{x}^{\frac{1}{2}}Nn_{x}$ is depicted in Table 5, whereas the output of Ω for $Re_{x}^{\frac{-1}{2}}Nn_{x}$ is not significant. Graphs of the dependent quantities of the velocity, temperature, concentration, and bioconvection profile are determined for three cases of mass transport (S=−0.5,0.0,0.5).

The role of incremented magnetic parameter *M* is to diminish the flow speed $f^{{}^{\prime }}\left(\eta \right)$. This is because the resistance force (Lorentz force) is enhanced with *M* to oppose the flow. However, with a rise in temperature θ(η), the fluid’s loss of kinetic energy is offset by a gain in heat energy, as indicated in Figure 2a,b. Figure 3a,b shows the porosity parameter $K_{p}$, on $f^{{}^{\prime }}\left(\eta \right)$ and θ(η), and *K*$\left(K_{P}=\frac{\nu }{K^{*}a}\right)$ is reciprocally against the permeability $K^{*}$ of the porous medium. The higher $K_{p}$ means lesser $K^{*}$, and therefore, higher Darcy resistance is provided to the flow to cause a decrement in the velocity and an increase in temperature. The drawings of $f^{\prime }\left(\eta \right)$ as an effect of progressive mixed convection parameter ω are shown in Figure 4a. Due to increasing buoyancy effects, the velocity increases directly with ω. Figure 4b delineates the reducing pattern of θ(η) when mixed convection parameter ϖ is improved. The larger convection transport absorbs more heat, and hence, the temperature decreases. Figure 5a,b and Figure 6a,b reveal the role of fluid material parameters β, δ on the velocity and temperature. When the parameter inputs are increased, the fluid velocity is faster and the temperature is lower. This is because these parameters are related inversely to viscosity. The higher β, δ means lesser viscosity, and hence, a faster flow results. When the flow speed increases, the temperature decreases.

For both the fluid and dust phases, Figure 7a,b depicts the nature of the local inertia parameter (Fr) on the velocity profile. The fluid velocity for both phases reduces with higher values of the local inertia parameter. The inertia coefficient is proportional to the medium’s porosity and drag force. As a result, as Cb increases, so does the porosity of the medium and the drag force. As a result, the liquid’s resistive force improves; as a result, the lower velocity attained corresponds to a higher local inertia value. The exact opposite behavior is delineated by the fluid temperature. Figure 8a,b demonstrates the role of the stretching/shrinking parameter ϵ on the velocity and temperature distribution. It can be seen that the higher values of ϵ improved the velocity profile. Physically, less shrinking increases the fluid velocity and decreases in the fluid temperature. Figure 9a shows the decelerating effect of aligned angle γ on the velocity profile f’(η). It can be seen that when the parameter’s γ inputs are increased, the speed of the flow for the shrinking surface increases. With the growing inputs of the Schmidt number Sc, the boundary layer concentration of the nanoparticles decreases, as portrayed in Figure 9b. The decline in concentration is observed for a large variety of diffusions. Figure 10a portrays the impact of Brownian motion Nb on the temperature profile θ(η): when the Brownian motion is increased, the fluid temperature also improves. Theoretically, the increase in the thermal conductivity of the nanofluid is mostly because of the increasing Brownian motion. It is observed from Figure 10b that for the improving values, thethermophoresis parameter Nt is higher. This is due to the thermophoresis impacting the transportation of nanoparticles from a higher degree to a lower degree, as the fluid temperature improves. Figure 11a shows the effect of Prandtl parameter Pr on temperature profile θ(η). Prandtl number Pr is equal to the quotient of the thick dispersion frequency to the thermal dispersion frequency, and a higher Prandtl number Pr lowers the thermal diffusivity. Figure 11b portrays the role of the bioconvection Lewis number on bioconvection profile χ(η). It can be seen that with higher values of Lb, the concentration of microorganisms decreases. Figure 12a shows the effect of Peclet number Pe on the bioconvection profile. As Pe increases, the concentration of microorganisms diminishes. The reduces the diffusivity of microorganisms, resulting in a decrease in the fluid’s motile density, and when density ratio Ω increases, the microorganism profile decreases, as observed in Figure 12b.

## 6. Conclusions

Work in this dissertation addressed the theoretical and computational analysis of bioconvection applications for magnetohydrodynamic Prandtl nanofluid flow caused by stretching in a horizontal sheet. The fluid flowed through a Darcy–Forchheimer porous medium. The non-dimensional boundary value problem was resolved numerically. The code of the Runge–Kutta method was developed in a MATLAB script. The major consequences of the influential parameters on the flow velocity volume fraction of the nanoparticle concentration of the microorganisms and the fluid temperature are concluded as below:The fluid velocity improved with improving ω, β, δ, ϵ, and γ, and it diminished against *M*, $K_{p}$, and Fr;The Temperature increased against the larger values of the thermophoresis parameters Nt and Brownian motion Nb, as well as the temperature rose with the increasing values of Kp and *M*, and it reduced when the mixed convection parameter ω improved;The concentration of the nanoparticles lowered when Sc rose;Graphs of the dependent quantities velocity, temperature, concentration, and bioconvection profile were determined for three cases of mass transport (S=−0.5,0.0,0.5);The microorganisms’ diffusion decreased when the values of the parameters Lb, Pe, and Ω increased;The skin friction was enhanced with $M,K_{p},\delta ,\beta ,Nr$, and Rb and diminished against ω;The results were compared with the past literature to validate the results;The Nusselt number decreased against Nb, Nt, and $S_{1}$, and it increased in magnitude directly with Pr;The increment in Pe and Lb resulted in a decline in the motile density number $Re_{x}^{\frac{1}{2}}Nn_{x}$.

## Figures and Tables

**Figure 1 nanomaterials-12-01791-f001:**
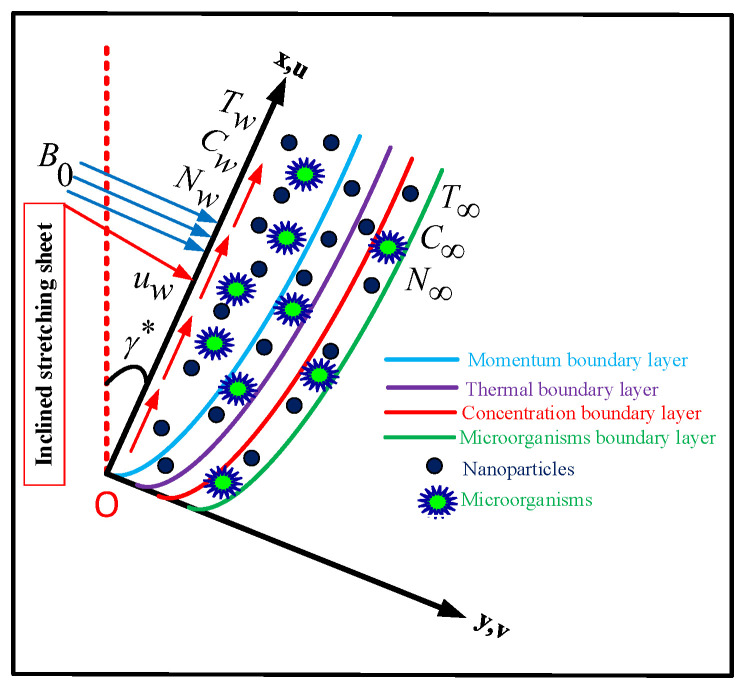
Physical representation of the problem.

**Figure 2 nanomaterials-12-01791-f002:**
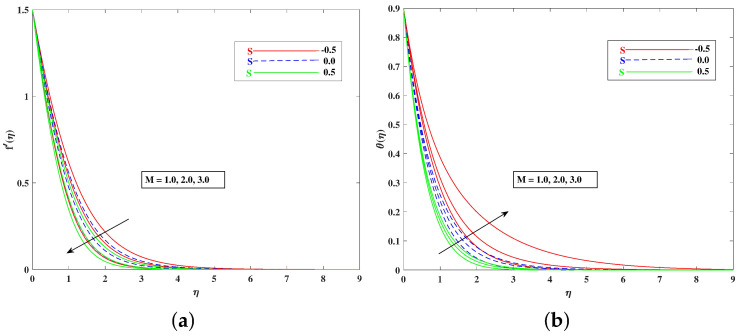
Effect of *M* on the velocity (**a**) and temperature (**b**) profile.

**Figure 3 nanomaterials-12-01791-f003:**
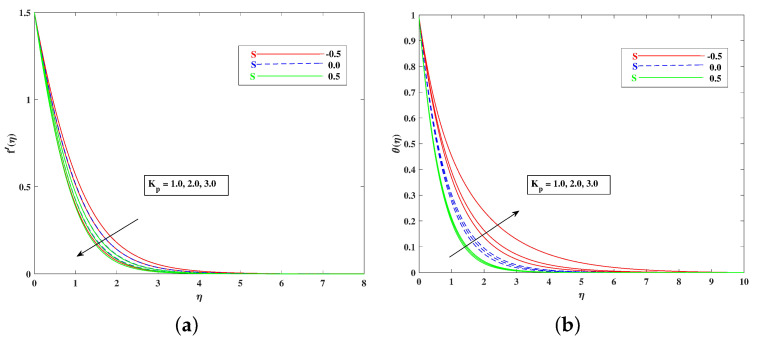
Effect of $K_{p}$ on the velocity (**a**) and temperature (**b**) profile.

**Figure 4 nanomaterials-12-01791-f004:**
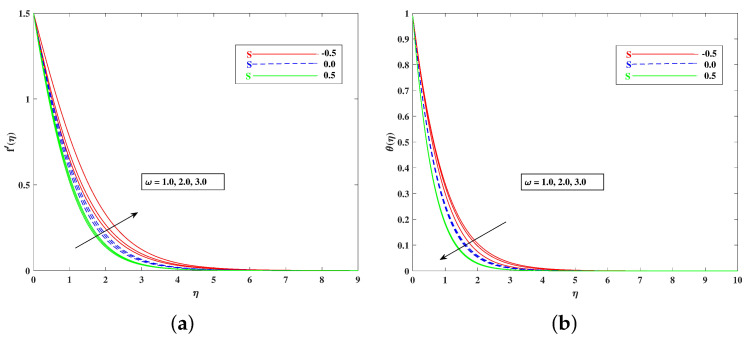
Effect of ω on the velocity (**a**) and temperature (**b**) profile.

**Figure 5 nanomaterials-12-01791-f005:**
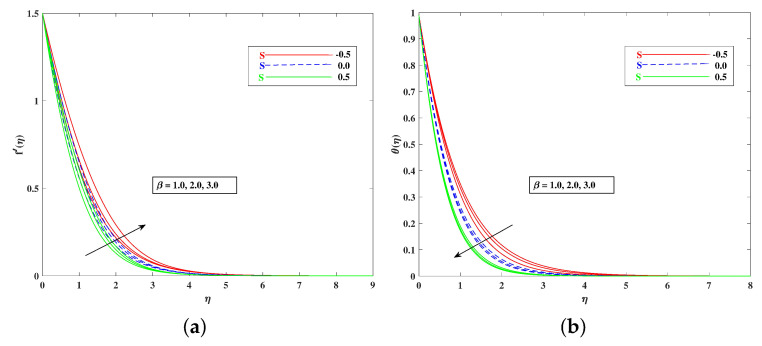
Effect of β on the velocity (**a**) and temperature (**b**) profile.

**Figure 6 nanomaterials-12-01791-f006:**
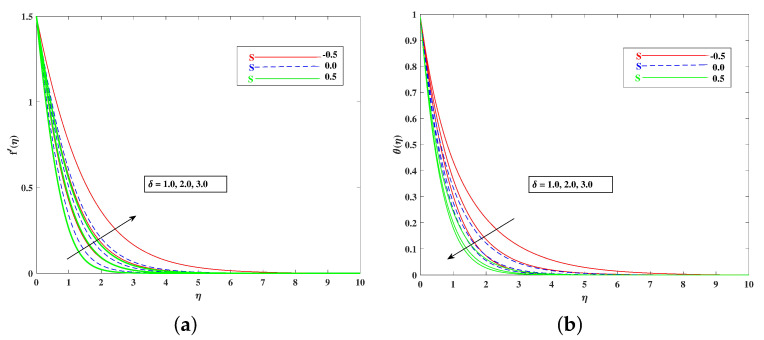
Effect of δ on the velocity (**a**) and temperature (**b**) profile.

**Figure 7 nanomaterials-12-01791-f007:**
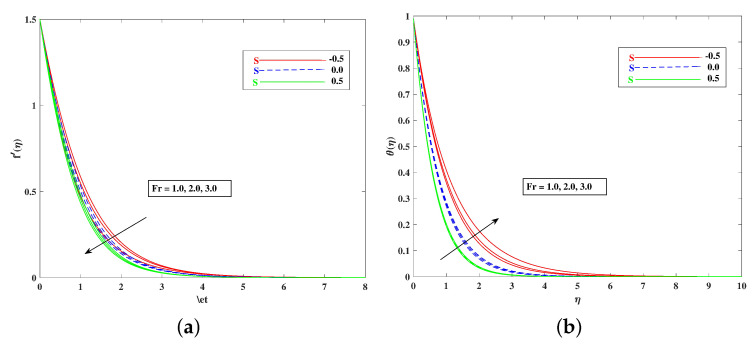
Effect of Fr on the velocity (**a**) and temperature (**b**) profile.

**Figure 8 nanomaterials-12-01791-f008:**
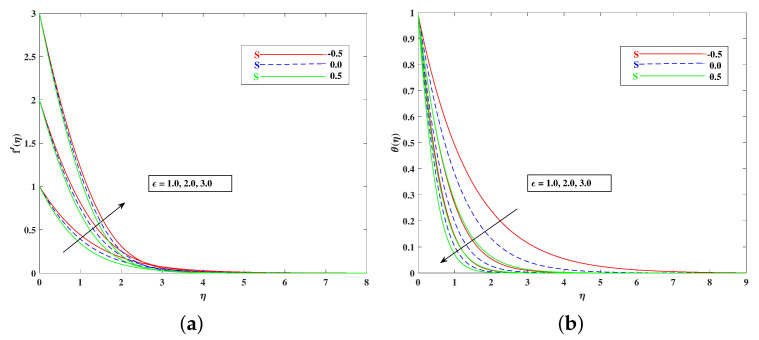
Effect of ϵ on the velocity (**a**) and temperature (**b**) profile.

**Figure 9 nanomaterials-12-01791-f009:**
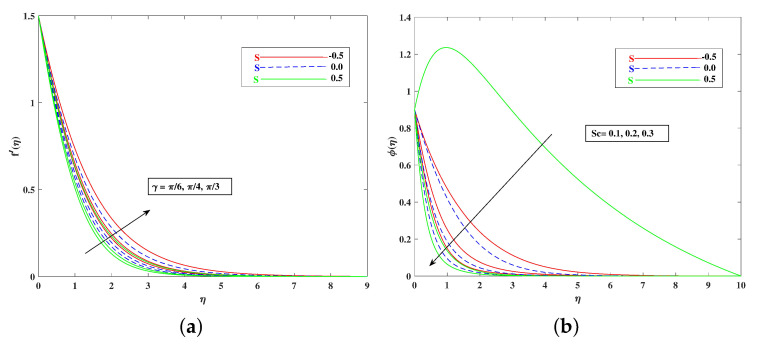
Effect of γ on the velocity (**a**) and Sc on concentration (**b**) profile.

**Figure 10 nanomaterials-12-01791-f010:**
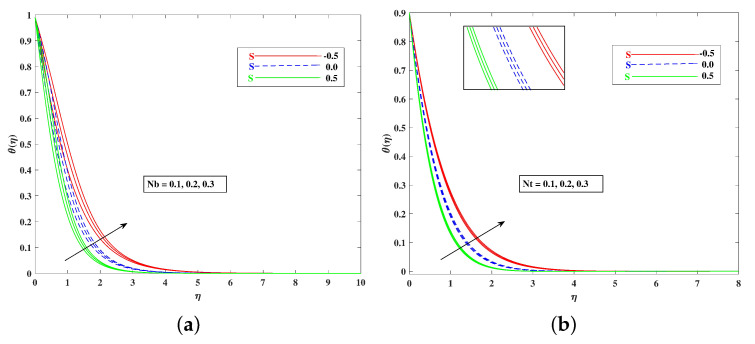
Effect of Nb and Nt on the velocity (**a**) and temperature (**b**) profile.

**Figure 11 nanomaterials-12-01791-f011:**
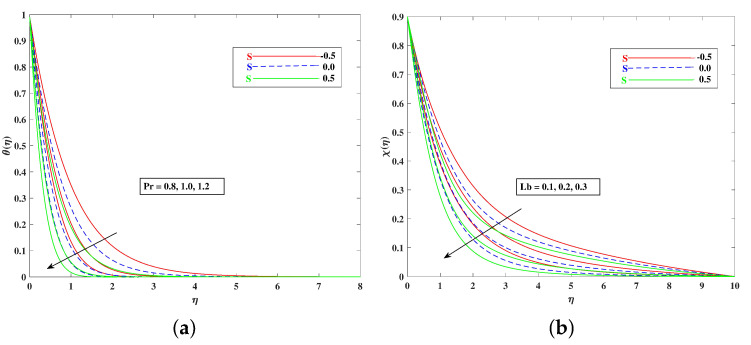
Effect of Pr and Lb on the temperature (**a**) and bioconvection (**b**) profile.

**Figure 12 nanomaterials-12-01791-f012:**
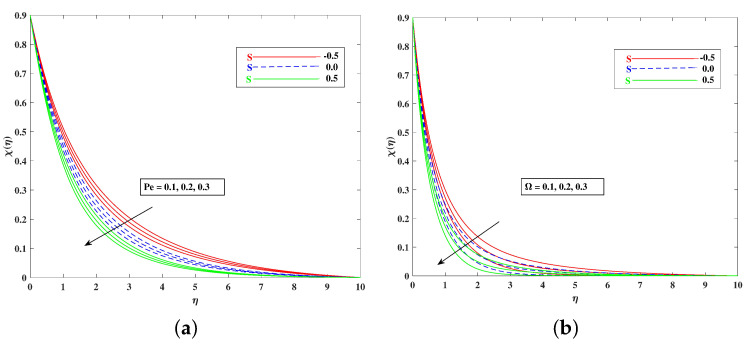
Effect of Peclet number Pe (**a**) and density ratio Ω (**b**) on the bioconvection profile.

**Table 1 nanomaterials-12-01791-t001:** Comparison of the values of ς when ϵ=1.0 (stretching case), Pr=1,M=0,S=0, and all other parameters are considered to be zero.

β	Abel [47]	Iskandar [48]	Bilal [43]	Our Results
0.0	−0.999962	−1.00000005	−1.0000000	−1.0000002
0.2	−1.051948	−1.05188989	−1.0518899	−1.0518896
0.4	−1.101850	−1.10190327	−1.1019044	−1.1019027
0.6	−1.150163	−1.15013734	−1.1501382	−1.1501488
0.8	−1.196692	−1.19671125	−1.1967134	−1.1967110
1.2	−1.285257	−1.28536326	−1.2863640	−1.2863740

**Table 2 nanomaterials-12-01791-t002:** Comparison of the values of *S* when ϵ=−1.0 (shrinking case), Pr=1,M=2,ς=0, and all other parameters are considered to be zero.

*S*	Bhattacharyya [49]	Iskandar [48]	Bilal [43]	(Our Results)
2.0	2.414300	2.41421357	2.41421369	2.41423
3.0	3.302750	3.30277563	3.30277621	3.30278
4.0	4.236099	4.23606797	4.23606814	4.23607

**Table 3 nanomaterials-12-01791-t003:** Impact of various physical parameters on the skin friction $Re_{x}^{\frac{1}{2}}C_{fx}=-\left(\frac{\delta \beta }{3}f^{\prime \prime 3}\left(0\right)+\delta f^{\prime \prime }\left(0\right)\right)$.

*M*	$K_{p}$	δ	β	Fr	ω	Nr	Rb	$C_{\mathrm{fx}}$
1.0	0.3	2.0	1.0	1.0	0.1	0.3	0.2	0.5154
2.0								0.6645
3.0								0.8036
	0.3							0.5154
	0.4							0.5234
	0.5							0.5314
		2.0						0.5154
		3.0						0.5954
		4.0						0.6641
			1.0					0.5154
			2.0					0.5159
			3.0					0.5163
				1.0				0.5154
				2.0				0.5214
				3.0				0.5274
					0.1			0.5154
					0.2			0.5037
					0.3			0.4843
						0.3		0.5154
						0.4		0.5537
						0.5		0.5922
							0.2	0.5154
							0.3	0.5188
							0.4	0.5223

**Table 4 nanomaterials-12-01791-t004:** Impact of various physical parameters on the Nusselt number $Re_{x}^{\frac{-1}{2}}Nu_{x}=\left(-\theta^{\prime }\left(0\right)\right)$.

Pr	Nb	Nt	$S_{1}$	${\mathrm{Nu}}_{x}$
0.8	0.1	0.1	0.3	0.5944
0.9				0.6647
1.0				0.7350
	0.1			0.5944
	0.2			0.5949
	0.3			0.5904
		0.1		0.5944
		0.2		0.5754
		0.3		0.5559
			0.3	0.5944
			0.4	0.5116
			0.5	0.4292

**Table 5 nanomaterials-12-01791-t005:** Impact of various physical parameters on the density of the motile microorganisms $Re_{x}^{\frac{-1}{2}}Nn_{x}=\left(-\chi^{\prime }\left(0\right)\right)$.

Pe	Lb	Ω	${\mathrm{Nn}}_{x}$
0.1	0.3	0.1	0.2838
0.2			0.3047
0.3			0.3258
	0.3		0.2838
	0.4		0.3246
	0.5		0.3593
		0.1	0.2838
		0.2	0.2853
		0.3	0.2868

## Data Availability

Not applicable.

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
