# Peer review of "Computational Analysis for Bioconvection of Microorganisms in Prandtl Nanofluid Darcy–Forchheimer Flow across an Inclined Sheet"

_nanomaterials, 2022, doi:10.3390/nano12111791_

Round 1
Reviewer 1 Report
Computational Analysis for Bioconvection of Microorgansims in Prandtl
Nanofluid Darcy-Forchheimer Flow Across an Inclined Sheet
The paper must be revised before the final acceptance:
- Firstly, the English language must be improved because you can see in the title, “Microorgansims” must be Microorganisms”
- What is meant by stratification in boundary layer flow?
- Right-hand side of equation (2) is not correct. Make the changes.
- Explain the Prandtl nanofluid, its physical significance, its all components of stress in detail. Bibliographic reference is not mandatory if you provide the right answer.
- At the end of the introduction, please expound on the new aspect of your work.
- Please utilize up-to-date and recent studies in the literature review to help the reader understand.
- Conclusions may be more specific and to the point; take a look and contemplate it.
- For those obtained results in this paper, authors should give some explanations in biology because they considered the bioconvection flow.
Author Response
Authors Response to the comments of Reviewer # 1
Dear respected Reviewer, We are thankful to you for your positive comments to improve our manuscript. We revised the manuscript according to the points and highlights the changes with color. Please look at the revised version.
Reviewer Comment# 1
Firstly, the English language must be improved because you can see in the title, “Microorgansims” must be Microorganisms”
Authors’ Response:
Done. Please look at the revised manuscript title. We update the title according to the wise suggestion
Reviewer Comment# 2
What is mean by stratification in boundary layer flow
Authors’ Response:
The notion of stratification is essential in lakes and ponds. It is important to control the temperature stratification and concentration differences of hydrogen and oxygen in such environments as they may directly affect the growth rate of all cultured species. Also, the analysis of thermal stratification is important for solar engineering because higher energy efficiency can be achieved with better
stratification. Researchers have shown that thermal stratification in energy storage may considerably increase system performance.
Reviewer Comment# 3
The right-hand side of equation (2) is not correct. Make the changes.
Authors’ Response:
Done. Please look at the revised manuscript. We update according to the wise suggestion
Reviewer Comment# 4
Explain the Prandtl nanofluid, its physical significance, its all components of stress in detail. Bibliographic reference is not mandatory if you provide the right answer.
Authors’ Response:
Done. Please look at the revised manuscript introduction.
Reviewer Comment# 5
At the end of the introduction, please expound on the new aspect of your work.
Authors’ Response:
Done. Please look at the revised manuscript's last paragraph of the introduction section.
Reviewer Comment# 6
Please utilize up-to-date and recent studies in the literature review to help the reader understand.
Authors’ Response:
Done. Please look at the revised manuscript introduction and references section. We update the manuscript according to the valuable suggestion.
Reviewer Comment# 7
Conclusions may be more specific and to the point; take a look and contemplate it.
Authors’ Response:
The conclusion section is modified as suggested.
Reviewer Comment# 8
For those obtained results in this paper, authors should give some explanations in biology because they considered the bioconvection flow.
Authors’ Response:
Done. Please see the introduction section which contains the paragraph about microorganisms.

Reviewer 2 Report
Authors have analyzed the bioconvection phenomena during Prandtl nanofluid flow over a stretching sheet. They have used Range-Kutta techniques to solve the system. The problem is well defined. However, I have some minor observations
- Include a nomenclature.
- Include some recent and relevant articles on the bioconvection and Darcy-Forchheimer flow model. Some of them are
International Journal of Biomathematics, (2021) 2150099, https://doi.org/10.1142/S1793524521500996 ; Journal of Heat Transfer - Trans of ASME, 143 (2021), 031201, https://doi.org/10.1115/1.4048946 ; International Journal of Ambient Energy, 2020, 1-19, https://doi.org/10.1080/01430750.2020.1818126 ; International Journal of Applied and Computational Mathematics, 5 (2019) 1-20, https://doi.org/10.1007/s40819-019-0705-0 ; Alexandria Engineering Journal, 60 (2021) 4067-4083, https://doi.org/10.1016/j.aej.2021.02.010 ; Heat Transfer, 51 (2022) 490-533, https://doi.org/10.1002/htj.22317.
- Figure 1 is not clear.
- The authors need to check the boundary condition (6) and need to explain how to get the equation (12) from (6).
- In fig 9(b), the figure for Sc = 0.1 does not converge quickly. Any reason behind it?
- Authors need to include the influence of Nn(x) for different parameters.
- References need to write in a proper format.
- The results and discussion part need to improve by including some physical significance in support of the nature of the graphs.
- Some typo errors or grammatical mistakes were observed in the paper.
Overall, the manuscript is well written. Graphs are adequate. The manuscript may accept for the publication after a minor revision.
Round 2
Reviewer 1 Report
Accepted
Author Response
We are thankful to you for your positive comments.